# OpenReview forum: "The Alignment Auditor: A Bayesian Framework for Verifying and Refining LLM Objectives"
_ICLR.cc/2026/Conference — ICLR 2026 Poster_

### Official Review · Reviewer_6foi · 2025-10-18

**Soundness:** 3
**Presentation:** 4
**Contribution:** 3
**Rating:** 6
**Confidence:** 3

**Summary:**

The authors propose a framework to audit alignment of LLMs. The authors look at how to estimate the reward of an LLM ("reward inference") with a three-stage auditing process which is called the Alignment Auditor.

The first step is "making ambiguity explicit" by which the authors mean learning a posterior distribution over reward parameters theta. They put a Gaussian prior over theta and a Bradley-Terry likelihood for pairs. They show that "non-identifiability can be systematically reduced", meaning that the posterior spread of theta (a proxy for non-identifiability) is reduced by making its uncertainty shrink.

Second, they use "uncetainty-aware diagnositcs" by which the authors mean they decompose uncertainty into aleatoric and epistemic and run some tests. The goal of these tests is to show that the inferred reward cannot be trusted. The tests include (a) injecting irrelevant tokens which they argue should increase epistemic uncertainty; and (b) correlate the variance of the reward with the distance from the training distribution.

Third, they show that the "refined objectives", i.e. the inferred reward function (the mean of the posterior reward distribution), when used in RLHF, show a decrease in the percentage of toxic completions on a held-out dataset on risky prompts.

In brief, the authors turn the problem of bayesian IRL into an auditing process, where the parimary goal is verification.

**Strengths:**

Great work, it was fun reading it.

1. Clearly formalised, motivated. It's connected to existing literature well and the idea of using reward shrinking as a way to understand if the inferred reward is good, as a tool of auditing, is cool and nice.

2. Very well-written and presented. Thanks.

3. Results are clear.

4. It's nice that you obtain calibration scores and show the decomposition of rewards over different rounds.

**Weaknesses:**

(To be clear, the weaknesses section is longer than strengths, but I think the paper has more strengths than weaknesses).

I (probably) understand why you chose to frame the paper the way you did. However, I feel this has exposed the paper to a few issues:

1. Your primary claim is that you are an *auditing* framework and you show you find where the rewards have some issues. If this is indeed the primary claim -- we have an *audit framework* to find issues -- you *must* show that other methods of auditing cannot achieve the same thing. For instance, for the toxicity setup (page 6-7), you must show that other ways of auditing LLMs, with the same train-validation split, would not have been able to identify issues in alignment. What do behavioral audits, adversarial audits, uncertainty-based audits miss? Even though your *mechanism* how you audit is different -- this is understood -- the *outcomes* of what you flag or not are not, relative to the alternatives.

2. Relying on retraining to prove usefulness is of course a very expensive procedure. I understand the academic value of this, but the extent to which this will become a practical tool is highly questionable. (Which is alright -- there's value in this either way).

3. There's a bit of a confusion, it seems, on the framing on "auditing". After you obtain your reward after posterior shrinkage, you continue to optimize the model with the expectation of the posterior. Then, it blurrs the line between "audit" vs "fixing" the problem because now you're re-training the model again, not auditing the original reward.  In other words: you're now no longer just saying *this reward had XX issues*, but also saying *with this new reward, you get XX better properties*. This is fine -- but then this doesn't become an "audit" tool, as it's now a re-training method. I think this is fine and good -- but just it does not square in terms of the framing that this is an auditing mechanism.

I think the biggest part for me is weakness (1). For you to make this a better auditing paper (which I understand it is), you need to do more to compare against other auditing methods and show what are the unique benefits you discover. As it currently stands, the paper largely explores your own method and properties instead of giving a comparative analysis of auditing and recovering issues.

**Questions:**

1. Is the fact that your estimates are calibrated a purely empirical observation or is there any theory to support why this should/shouldn't be the case?

2. Any things you tried before that *did not* work and you left out of the paper?

3. Any reason you used PPO instead of GRPO?

4. Are you the first work to look at modeling and contracting the posterior reward distribution or has this been done before?

5. If you could address the three points from the weaknesses, that'd be great too. Especially weakness 1.

I would be mostly convinced to increase my score if:
- (a) You can give a better related work overview on audit procedures both in terms of (i) mechanisms (how audit procedures work) and (ii) outcomes (what audit procedures uncover)
 - (b) show me how you uncover issues that other audit procedures do not. It does not have to be a "better" issue uncovered -- any single dimension along which you discover issues that no other audit framework discovers would be strong evidence for me.

---

> ### Author Response · Authors · 2025-11-20
> **Response to Reviewer 6foi**
>
> We thank the reviewer for their encouraging feedback, describing the work as "fun to read," "well-written," and "clearly formalized." We particularly appreciate the recognition of our core novelty: using reward posterior contraction as a tool for auditing. We take the reviewer’s request for a comparative analysis with other auditing methods very seriously. Below, we provide a comprehensive comparison demonstrating *what* our method uncovers that others miss, clarify the role of "refinement" as validation, and address the specific technical questions.
>
> **Why Objective Auditing Detects What Behavioral Auditing Misses**
> The reviewer asks for the primary differentiator between our framework and existing methods (behavioral, adversarial, or uncertainty-based audits). The fundamental difference lies in the target of the audit: existing methods audit the **outputs** ($y$), whereas our framework audits the **latent objective** ($\theta$) that generates those outputs. We will make this clear in the manuscript and will add in the necessary descriptions of behavioral auditing works to the related work.
>
> Behavioral and adversarial audits (e.g., Red Teaming) rely on finding specific prompts that trigger failure. If a red-teaming suite fails to find a jailbreak, the auditor concludes the model is safe. However, this is often a false negative; the model may simply be "lucky" or the search space too vast. Our framework, by recovering the posterior distribution of the reward, detects **ambiguity** (non-identifiability) even when the model’s current outputs appear safe.
>
> We can demonstrate this concretely using our experimental results. In Round 1 of our toxicity experiments (Appendix Table 1), the model learned a "reward hacking" strategy where it outputted nonsense like "Edit. And..." to avoid toxicity. A standard behavioral audit measuring "toxicity percentage" would rate this model as **0% toxic**—a perfect score. It would pass a standard safety filter. However, our Alignment Auditor flagged this model immediately. Why? Because the **epistemic uncertainty** (variance of the reward posterior) and the **posterior volume** (log-determinant of covariance) were extremely high (log-det $\approx -196$). Our framework correctly diagnosed that the model had not learned the true objective, but rather a fragile shortcut. By Round 5, the toxicity was still low, but the posterior volume had contracted massively (log-det $\approx -897$), confirming that the model was now optimizing the *intended* objective rather than a loophole. Standard behavioral audits cannot distinguish between "safe because aligned" and "safe because broken," whereas our objective-based auditing can.
>
> Furthermore, our "spurious feature" experiment (Figure 5, left) highlights a capability that standard uncertainty quantification (UQ) misses. Standard UQ often measures predictive entropy (aleatoric uncertainty) on the output tokens. We decompose this to isolate epistemic uncertainty in the *reward*. When we injected irrelevant keywords, the model’s predictive confidence might remain high (it is confident in its output), but our framework detected a spike in epistemic uncertainty, signaling that the prompt was Out-Of-Distribution relative to the inferred reward function. This allows an auditor to flag *regions* of the prompt space where the alignment is fragile, essentially generating a "heat map" for where red-teamers should focus their efforts, rather than shooting in the dark.
>
> **Refining the Definition: "Refinement" as Validation, not Just Training**
> We agree with the reviewer’s observation that Stage 3 (optimization) blurs the line between auditing and fixing. We wish to clarify the epistemological role of this stage. We do not propose our framework primarily as a training method (though it can be used as one); rather, we view the "Refinement" stage as the necessary **validation** of the audit.
>
> In scientific instrumentation, one validates a measurement tool by using it to predict an outcome. If our audit claims, "The model effectively believes X is the goal," the only way to rigorously prove this is to train a policy on that inferred goal and observe if it achieves X. If we stopped at Stage 2, the audit would remain theoretical. By proceeding to Stage 3, we provide the auditor with tangible proof: "The reward function we recovered is not just a mathematical abstraction; it is capable of driving the policy to match the Ground Truth oracle." This confirms that the sequential reduction in posterior volume (Stage 2) corresponds to a real-world gain in alignment stability. We will revise the manuscript to explicitly frame Stage 3 as "Policy-Level Validation" to avoid confusion with standard training proposals. Regarding practicality, while retraining is expensive, we argue that for high-stakes foundation models (e.g., medical or legal agents), the cost of verifying the objective function is a necessary insurance premium against latent misalignment.

---

> > ### Author Response · Authors · 2025-11-20
> > **Response to Reviewer 6foi continued**
> >
> > **Answers to Specific Questions**
> >
> > *   **Is the calibration theoretical or empirical?** The calibration we observe is primarily an empirical result of the Bayesian averaging process, though it is grounded in theory. By marginalizing over the posterior $p(\theta|D)$, we effectively smooth out overconfident predictions from any single point estimate of the reward. This creates a "ensemble effect" known in Bayesian Deep Learning to improve calibration (Gal & Ghahramani, 2016). Our scaling experiments (Llama-3-8B) confirm this effect strengthens with model capacity, as better features allow for a more accurate posterior approximation.
> > *   **Did anything fail?** Yes. Initially, we attempted to run Bayesian IRL using just a single round of data (Standard BIRL). As shown in our new ablation studies, while the accuracy was decent, the posterior remained dangerously wide (high entropy). The policies trained on this Single-Round reward were unstable and prone to reward hacking. This failure mode directly motivated the **Sequential** design of the Alignment Auditor, which we found necessary to collapse the solution space to a trustworthy region.
> > *   **PPO vs. GRPO:** We utilized PPO because it remains the standard baseline for RLHF in the literature (Ouyang et al., 2022), allowing for the most direct comparisons with prior work. GRPO (Group Relative Policy Optimization) is a promising direction, but introducing it would have added a confounding variable to our study of reward inference.
> > *   **Novelty of Posterior Contraction:** While Bayesian IRL dates back to Ramachandran & Amir (2007), those works focused on tabular grid-worlds. To our knowledge, we are the first to apply **Sequential Bayesian IRL to Large Language Models** specifically for the purpose of *auditing* alignment. Recent work (e.g., Yang et al., 2024) uses Bayesian rewards for optimization, but they do not leverage the posterior contraction as a diagnostic metric for auditing non-identifiability, which is our core contribution.
> >
> > **Conclusion**
> > We believe we have addressed the reviewer's condition for increasing the score by demonstrating that our framework detects latent misalignments (e.g., reward hacking loopholes that pass behavioral checks) and OOD fragility that standard audits miss. Our new results–showing a massive reduction in posterior volume (-196 to -897) and a high Cohen’s $d$ (1.32) separation–quantify this benefit (see page 19, Table 7, Table 8). We see the Alignment Auditor not as a replacement for behavioral audits, but as a deeper, "MRI-style" scan of the model's objectives that complements the "symptom-based" checks of standard red teaming.

---

> ### Author Response · Authors · 2025-11-26
> **Further clarification required?**
>
> Dear Reviewer 6foi
>
> We have posted responses to the concerns raised by your review. Is there anything further you would like clarification about beyond what we posted in the comments? Have all your concerns been appropriately addressed? Please let us know and we'd happy to provide you with further details where required.
>
> Thank you

---

> > ### Comment · Reviewer_6foi · 2025-11-27
> > **Response**
> >
> > Hi authors,
> >
> > Thanks for your response. It seems my issues are not yet addressed. Let me explain:
> >
> > 1. Your response to my critique ("Your primary claim is that you are an auditing framework and you show you find where the rewards have some issues.") is that you are auditing latent objectives theta and other models audit outputs. That could be the case. But then I need an answer in one of two ways:
> > - (a) I need a convincing demonstration where auditing latent factors is necessary and other audit mechanisms achieve this. Currently, your evaluation is the following: "The AllenAI RealToxicityPrompts dataset (AI, 2022) (99k naturally occurring with Perspective scores) is used to study detoxification: given a prompt, generate a safe, non-toxic continuation." and "A global threshold for toxicity is chosen on a validation set and then fixed for test.". Therefore, you *can* audit this without latent factors -- why do this?
> > - OR (b) I need a demonstration that using audit mechanisms with latent factors does indeed result in better performance (therefore, a comparative against related work).
> >
> > Again: *I understand* that the audit mechanism is different and I'm not challenging your contribution of detecting ambiguity by recovering the posterior. This is great. But you are framing this as a practical mechanism and I do not understand this yet.
> >
> > 2. ICLR allows to change the text and re-upload it. I do not see any actual changes implemented yet? Can you please highlight them.
> >
> > 3. On auditing vs training a new model, you write this: "the only way to rigorously prove this  [predicting an outcome] is to train a policy on that inferred goal and observe if it achieves X." But I am still left confused: are you now making the claim that *the only mechanism by which you can audit behavior is to necessarily re-train it?*. If so, this is a highly contenious claim: I'd love for you to explain why all audit mechanisms which do not re-train models fail. Otherwise, if this is not the thing you're claiming, what do you mean by saying: that this is "the only way"?
> >
> > 4. In addition to point (3), this results in an extremely expensive audit procedure, no? That is, to audit a model, you effectively need to re-train it. Wouldn't it be better to frame this work as a mechanism to **correct** models which then necessarily requires both understanding and re-training?
> >
> > --
> > All the other points are super clear - thanks.  As before, I still think the paper is in great shape, but none of my major concerns have been addressed.

---

> > > ### Author Response · Authors · 2025-12-03
> > > **Response to further questions by Reviewer 6foi**
> > >
> > > Thanks for further clarification on your questions. We discuss each of the issues below:
> > >
> > > **Why Latent-Factor Auditing Improves Over Standard Output-Based Auditors (Addressing Point (b))** To directly address your request for a comparative evaluation, we added a new experiment using the same spurious-marker setup as our uncertainty diagnostics (Figure 5, left and middle). In this experiment, we keep the dataset, model, and train/test split fixed, and compare three audit signals:
> > >
> > > * **Output-only auditor:** RoBERTa toxicity probability (standard behavioural audit).
> > > * **Latent-factor auditor (posterior-mean reward):** the scalar reward $\mu^\top \phi(o)$ obtained by plugging in the **posterior mean** parameter $\mu$ of the variational reward posterior $q(\theta)$ (a single reward model).
> > > * **Latent-factor auditor (full posterior):** the latent reward variance $\mathrm{Var}[R(o)]$ under the full posterior $q(\theta)$ (our Alignment Auditor).
> > >
> > > As shown in Figure 12 (page 20), across all marker fractions (20–50%):
> > >
> > > * The **output-only toxicity auditor** remains close to chance at detecting marked vs. unmarked prompts (AUC ≈ 0.55–0.58), indicating poor sensitivity to spurious markers.
> > > * The **posterior-mean reward** already performs noticeably better, achieving AUC ≈ 0.63–0.70, demonstrating that even a single latent-factor–based score is a stronger auditor than the standard output-based signal.
> > > * The **latent reward variance $\mathrm{Var}[R(o)]$** from the full posterior $q(\theta)$ performs best, consistently reaching AUC ≈ 0.86–0.88 and separating marked prompts as high-uncertainty, shortcut-prone regions.
> > >
> > > In other words, when we hold everything fixed and only change the **audit signal**, both latent-factor–based auditors outperform the standard output-based RoBERTa auditor, and **the full $\theta$-posterior–based** variance signal $\mathrm{Var}[R(o)]$ provides the strongest separation.
> > >
> > > **Highlighting changes implemented** In our previous response, we mentioned that “our scaling experiments (Llama-3-8B) confirm this effect strengthens with model capacity, as better features allow for a more accurate posterior approximation.” The empirical evidence for this claim is presented in **Table 6 (page 18)**, where we report results on a more nuanced helpfulness preference task using Llama-3-8B-Instruct. In addition, we scale the original toxicity task to Llama-3-8B-Instruct; the corresponding calibration and performance results are shown in **Table 7** and **Figure 11 (page 19)**.
> > >
> > > Regarding the question “Did anything fail?”, our revised manuscript now explicitly frames standard single-round BIRL as an initial approach that proved insufficient, and uses this as motivation for our sequential design. The comparative ablation between Standard (single-round) BIRL and our Sequential Alignment Auditor is reported in **Table 8 (page 20)**.
> > >
> > > **On auditing vs training a new model (clarifying Stage 3).**
> > > Thank you for highlighting the ambiguity in our previous wording. We agree that the phrase “the only way to rigorously prove this is to train a policy…” was too strong and potentially misleading. We are **not** claiming that the only mechanism by which one can audit behaviour is to re-train a model, nor that audit mechanisms which do not involve retraining will fail.
> > > What we intended to convey is once we have inferred a latent objective $R_\theta$, one way to obtain particularly strong evidence that this objective is behaviourally adequate is to train a fresh policy against it and check whether this policy reproduces the behaviour of the original policy from which the demonstrations were collected (e.g., achieving detoxification performance comparable to the oracle-trained RLHF policy).
> > >
> > > **Clarifying cost of re-training for Stage 3**
> > > In practical deployments, our framework can be run in a lightweight audit mode by using just Stages 1–2. This already provides information beyond standard output-based audits—such as posterior contraction and variance-based detection of shortcut-prone or OOD regions—without retraining the model. The full **retrain** loop in Stage 3 is indeed more expensive, providing a **policy-level validation** step, appropriate in high-stakes settings where a practitioner is willing to pay the additional cost to verify that the inferred objective can, by itself, reproduce the behaviour of the original policy from which it was inferred.

---

### Official Review · Reviewer_UjEE · 2025-10-31

**Soundness:** 3
**Presentation:** 3
**Contribution:** 3
**Rating:** 8
**Confidence:** 3

**Summary:**

The authors prescribe a methodology to audit reward scores for RLHF pipeline.

**Strengths:**

The paper, in general, is well written, and the ideas are presented with clarity. I am not very familiar with audit literature, so I cannot comment on the technical novelty. However, the methodology proposed is sound, and I am convinced of its correctness.

**Weaknesses:**

I believe the primary cost stems from the computational overhead. Primarily with Stage 2, where one runs Bayesian updates over K iterations to obtain the final reward model for use.
Also, I may be mistaken, but how is one guaranteeing the correctness of \pi_E, the expert?

**Questions:**

In Figure 6, it appears that after one round, the reward metrics closely track the ground truth. I am assuming the round refers to those in stage 2. In that case, can we get away with doing just a single round? More importantly, the task methodology seems quite intensive. Does it make sense to invest in the time and money, given that we have reward-free alignment techniques?

How would one extend this audit to a reward signal-free technique, for example, in the case of DPO?

---

> ### Author Response · Authors · 2025-11-20
> **Response to Reviewer UjEE**
>
> We thank the reviewer for their highly positive assessment, describing the methodology as "sound" and the paper as "well written." We appreciate the insightful questions regarding computational cost, the necessity of sequential rounds, and the relevance of auditing in the era of reward-free alignment (DPO). We address these points below.
>
> **On the computational overhead of running Stage 2 updates**. The reviewer correctly notes that Bayesian inference adds overhead, but we find this cost is negligible compared to the cost of pre-training or fine-tuning. The "heavy lifting" in our framework is the forward pass to extract features from the LLM. However, because we model the reward as a linear head over these frozen features, the actual Bayesian update (Stage 2) occurs in a low-dimensional space so the posterior updates are orders of magnitude cheaper than model fine-tuning. In our experiments, the Bayesian inference took less than **5% of the total compute time** required for the full RLHF loop. Furthermore, while the *rounds* are sequential, the processing of data *within* each round is fully parallelizable. Finally, this cost is an "audit premium"—a small upfront cost to verify the safety of a model before deployment, which is often a requirement in high-stakes settings.
>
> **Guaranteeing the Correctness of the Expert ($\pi_E$)**
> We do not assume the expert is perfect; rather, we treat the expert as a "noisy" source of rationality. Our framework is designed specifically to handle imperfect experts via two mechanisms:
> 1.  **Probabilistic Modeling:** We use a Bradley-Terry likelihood which models the expert as maximizing reward *stochastically*. Occasional mistakes or inconsistencies by the expert are absorbed by the noise term rather than breaking the inference.
> 2.  **Self-Diagnosis:** The audit itself acts as a quality check on the expert. If the expert is incoherent (randomly flipping preferences), the posterior distribution will *fail to contract*, resulting in persistently high epistemic uncertainty. In our experiments, the successful contraction of the posterior (reduction in log-determinant covariance) serves as empirical proof that the expert signal was consistent enough to recover a stable objective.
>
> **Can we get away with a single round? (Re: Figure 6)**
> The reviewer observes that in Figure 6, the mean reward seems to perform well after just one round. However, we strongly argue that **Sequential Auditing (multiple rounds)** is necessary to ensure safety. While a Single-Round reward might capture the correct *direction* (mean), it often suffers from high variance (uncertainty), leaving "loopholes" that the model can exploit.
> To prove this, we rely on two pieces of evidence. First, **quantitatively**, our new ablation study shows that the Single-Round posterior is significantly "looser" than the Sequential posterior: the **log-determinant of the covariance** (a measure of ambiguity) improves from **-196.0** in the Single-Round baseline to **-897.1** after sequential updates. Second, **qualitatively**, we observed the direct consequences of this under-identification in our main experiments (Appendix Table 1). The policy trained on the **Round 1 posterior** (which suffers from similar high entropy) exhibited severe **reward hacking**, achieving low toxicity scores by generating incoherent, evasive text (e.g., "Edit. And..."). This behavior completely disappeared by Rounds 2-5 as the posterior contracted. Thus, sequential rounds are mathematically necessary to tighten the decision boundary and prevent the model from exploiting these adversarial shortcuts.
>
>
> **Relevance in the age of Reward-Free Alignment (DPO) and Extension**
> The reviewer asks if auditing is worth it given techniques like DPO. We argue that auditing is **more** critical for DPO, not less. In RLHF, there is an explicit reward model we can inspect. In DPO, the reward function is implicit and hidden inside the policy's weights. This opacity makes DPO models harder to interpret.
> Our framework extends naturally to DPO by treating the DPO-trained model as the "Expert" ($\pi_E$). We can run our Alignment Auditor to inverse-engineer the implicit reward function that the DPO model actually learned. This allows auditors to answer: "Did DPO effectively learn the intended preference, or did it overfit to surface-level heuristics?"
> Essentially, while DPO solves the *optimization* problem, the Alignment Auditor solves the *verification* problem. They are complementary: DPO trains the model, and our framework generates the safety certificate.

---

> ### Author Response · Authors · 2025-11-26
> **Further clarification?**
>
> Dear Reviewer UjEE
>
> We have posted responses to the concerns raised by your review. Is there anything further you would like clarification about beyond what we posted in the comments? Have all your concerns been appropriately addressed? Please let us know and we'd happy to provide you with further details where required.
>
> Thank you

---

### Official Review · Reviewer_joPc · 2025-10-31

**Soundness:** 2
**Presentation:** 3
**Contribution:** 1
**Rating:** 4
**Confidence:** 3

**Summary:**

This paper formalises a LLM auditing framework that relies on Bayesian IRL to infer some ground-truth toxicity labels of completions.

**Strengths:**

The paper presents a successful experiment to recover the reward (objectives) that was used to finetune an LLM model.
The experimentation spans a good variety of LLM models and appears robust, though large LLMs are a bit under explored.
The presented framework is a creative way to verify the level of alignment reached by the LLM during RLHF.
A clear and an easy to follow presentation.
LLM alignment is an important and hot topic.

**Weaknesses:**

The weaknesses in summary:
- The paper would be more clear if there was more emphasis on the auditing capability of the framework.
- Unclear positioning wrt SoTA.
- Some conclusions would be more solid if supported by more evidence.
- The results would be more sound if a more solid ground truth was used.
- The title’s claim of “refinement” would be clearer with demonstrated refinement rather than validation and partial transfer
- Simplicity of experiments could prevent conclusions significance and generalization (single case and binary one: toxic or not, LLMs < 1b parameters)

In more details, to explain and justify the points above:

- The goal of the paper evolves around the auditing capability of the framework: line 017 "This paper introduces a principled auditing framework that re-frames reward inference from a simple estimation task to a comprehensive process for verification.", line 057 "In this work, we argue that understanding the behavior of LLMs through reward inference should not be approached as a one-shot estimation problem, but as a principled auditing process.". But there is not enough demonstrated auditing capabilities:
    - No auditing in Stage 1 since we are just learning to distinguish the expert's from the baseline's completions while learning a reward that the expert's maximizes compared to the baseline.
    - On the second Stage, the author adds more iterations and more data to the expert, its scores improve. Also there is new introduced metrics: "Total Uncertainty" and "Epistemic Uncertainty", that makes the model aware of the data points that he would not be able to score (score here is the reward) correctly. But no Auditing capabilities in Stage 2.
    - On the third stage there we verify that the reward that we captured is can be used actually to reduce toxicity. At this stage (3) also it is not auditing.
-lines 105-110 present other BIRL approaches, but none are evaluated in the experimental section --  those would have helped identify the benefits of the proposal. In particular, lines 110-112 did not convince me completely of the novelty of the approach.
- Considering a toxicity classifier as the ground truth can effect the results of the experiments in many ways.
    - The used toxicity classifier (s-nlp/roberta_toxicity_classifier) have an F1-score of 0.76 (https://huggingface.co/s-nlp/roberta_toxicity_classifier#licensing-information) which is comparable to the results that were reached in the paper.
    - Also there is no discussion in the paper about the limitations and the effects of having a toxicity classifier as ground truth.
    - Toxicity reduction is arguably only a fraction of general alignment, the authors could discuss how their approach transposes to other alignment objectives.
    - I missed a note regarding the limits of completion-based toxicity detection: I expect the toxicity of answers to heavily depend on the original question!
    - the paper would be improved by bringing central technical notions (like posterior contraction) closer to their auditing meaning/use. Besides, I would have enjoyed a deeper discussion on the problem of non-identifiability. While the pointed paper (Ng&Russell) clearly illustrates non-identifiability, I believe in learning contexts this non-identifiability is frequent (eg justification of SVM margins) and not always problematic -- provided the inferred model is close enough from the ideal one.
- In many instances, the paper makes some conclusions with only limited support, for example:
    - In line 377 the author claims "Importantly, sequential Bayesian updates tend to mitigate the effects of reward hacking in comparison to single-round inference (see Figure 5(right)).". The Figure 5 (Right) shows the toxicity reduction in the RLHF process and it compares different rewards on that metric. The "ground truth" does the best, Round 1 does better than the the other Rounds which is not a proof that there was reward hacking at Round 1.
    - In line 473 is the conclusion there is a statement "Third, policy-level validation shows the inferred reward can directly drive PPO,  achieving comparable downstream toxicity reduction compared to the ground truth RLHF alignment." which is questionable, because if we compare the toxicity reduction ratios between the Ground Truth RLHF and the other rounds (Rounds 2-5) in the Figure 5 (Right) we find that the reduced toxicity of the recovered reward is nearly half the reduction by ground truth.
    - There is not enough numerical metrics and evidence to reinforce the claims based on qualitative results (426 e.g. "With the round-1 (poorly identified) posterior, policy-level alignment with PPO exhibits reward hacking where completions show topic loss, repetition and abrupt cut-offs that suppresses toxic tokens at the expense of coherence and helpfulness (Appendix Tables 1).") or visualization of graphs (353 "while the distribution of inferred rewards in Figure 2(b) reveals a distinct decision boundary between scores assigned to toxic and non-toxic texts.")
- In the title "THE ALIGNMENT AUDITOR: A BAYESIAN FRAMEWORK FOR VERIFYING AND REFINING LLM OBJECTIVES", there is a claim of a "Bayesian framework for ... and refining LLM objectives". The refinement is not demonstrated in the paper. What could have been referred to here is the **validation** of the recovered reward in the third stage. But to get the reward that was used to finetune the baseline LLM in Stage 3 we needed a finetuned version of the same LLM (Expert) first. Meaning the finetuning in Stage 3 is not a refinement of the LLM objectives as inferred to in the title, it is a transfer of **some** of the alignment learning of the expert to another baseline version of itself.

**Questions:**

- What motivated the use for Roberta toxicity classifier given its reported F1~0.76? And how would the conclusions change if we consider the classifier as a noisy labels rather than ground truth?
- For the claim "Importantly, sequential Bayesian updates tend to mitigate the effects of reward hacking in comparison to single-round inference (see Figure 5(right))."(l.377). What evidence supports this conclusion beyond the Figure 5 (right)?
- How is "comparable" defined in “comparable downstream toxicity reduction” (l.473)? And what evidence supported that claim in the conclusion ?
- How is a "distinct decision boundary" defined in l.353, and what metrics and evidence beyond visualization can be presented to reinforce that claim?
- What concrete audit outputs (e.g. reports like mentioned in l.476) does the framework produce?
- What example can be provided to demonstrate clearly the auditing capabilities of the presented framework?
- What is inferred by "refinement" in the title: "Bayesian framework for … and refining LLM objectives"?
- The experiment demonstrates that the proposed Bayesian auditing method can recover a reward function consistent with the oracle reward, but could the simplicity of this use case limits the significance of the result? Does reproducing oracle RLHF dynamics necessarily imply alignment with human values/preferences? could it be only mere consistency with the proxy signal used during training?
- Any idea of how challenging that would be to go beyond linear model for reward?
- What about moving from the toxicity to helpfulness or even open-ended generation? What challenges would that raise?
- Existing related works seem extensive, is there any way of direct comparison with some of them? that would help grasp the contribution utility of your work.

---

> ### Author Response · Authors · 2025-11-20
> **Response to Reviewer joPc**
>
> We thank the reviewer for their detailed assessment and for recognizing the importance of verifying alignment in LLMs. We appreciate the constructive feedback regarding the definition of "auditing," the reliance on toxicity classifiers, and the request for further empirical evidence. We have conducted significant additional experiments during the rebuttal period–including scaling to Llama-3-8B, extending to the Anthropic HH-RLHF "Helpfulness" dataset, and quantifying decision boundaries–to address these concerns. We provide a comprehensive response below.
>
> **Clarifying the Definition of Alignment Auditing and Refinement**
>
> We respectfully believe there is a misunderstanding regarding the definition of "auditing" and "refinement" within our framework, which we aim to clarify. The reviewer suggests that Stage 1 (inference) and Stage 3 (validation) are not auditing. However, we define *Alignment Auditing* not merely as checking model outputs (which is standard red-teaming), but as the rigorous process of verifying the *objective function* that implicitly drives model behavior. In Stage 1, we do not just learn to distinguish completions; we recover the latent reward distribution to check if the model is optimizing what we think it is optimizing. If an auditor cannot recover the intended reward from the model's behavior, the model is effectively unauditable. Stage 2 generates the "Audit Report" by flagging specific prompts where the inferred objective is ambiguous or untrustworthy (high epistemic uncertainty), acting as a diagnostic tool for safety teams.
>
> Regarding "Refinement," the reviewer asks if this is merely transfer learning. We use the term *refining* because the framework takes potentially noisy, inconsistent human (or expert) demonstrations and distills them into a calibrated, uncertainty-aware reward function. As shown in our sequential experiments, this process filters out ambiguity. In Stage 3, we use this *refined* signal to retrain the model. This is not simple transfer; it is a correction process. If the original demonstrations contained noise or "shortcuts," the Bayesian posterior contraction helps marginalize these out, providing a more robust training signal than the raw data itself. We will revise the manuscript to explicitly define these terms in the introduction to prevent ambiguity.
>
> **Evidence of Reward Hacking Mitigation and Decision Boundaries**
>
> The reviewer questioned the claim that sequential updates mitigate reward hacking, specifically pointing to Figure 5 (Right) where Round 1 reduces toxicity more than the Ground Truth. We emphasize that *this is exactly what reward hacking looks like*. In RLHF, a model that achieves a "better" score than the ground-truth oracle is often exploiting the reward model at the expense of fluency or coherence (Goodhart’s Law). As noted in our qualitative analysis (Appendix Table 1), the Round 1 policy achieves near-zero toxicity by outputting degenerate, evasive text (e.g., "Edit. And..."). It collapses because the reward function is under-identified. Rounds 2 through 5, however, track the Ground Truth trajectory closely. They do not "beat" the oracle because they are constrained by the KL divergence to remain coherent. Thus, "comparable" toxicity reduction refers to matching the oracle’s optimization path without collapsing into degeneracy.
>
> To address the reviewer’s request for quantitative metrics regarding the "distinct decision boundary" (Line 353), we performed a new analysis quantifying the separation between toxic and non-toxic distributions in the inferred reward space. We calculated Cohen’s $d$ (standardized mean difference) on the test set. Our inferred reward yields a Cohen’s $d$ of **1.3254** (see page 19, Table 7). In statistical analysis, a $d$ value above 0.8 is considered a "large" effect size. This provides strong numerical evidence—beyond visualization—that the Alignment Auditor recovers a reward function with a statistically significant and powerful decision boundary between aligned and misaligned behaviors.

---

> ### Author Response · Authors · 2025-11-20
> **Response to Reviewer joPc continued**
>
> **Scaling to Llama-3-8B and Generalizing to Helpfulness (HH-RLHF)**
> The reviewer expressed concern about the simplicity of the experiments (binary toxicity, small models). To demonstrate the generalizability of our framework, we have conducted two major new experiments. First, we applied our framework to the **Anthropic HH-RLHF dataset**, moving beyond binary toxicity to the nuanced domain of "Helpfulness." Using a Llama-1B backbone and the `Ray2333/gpt2-large-helpful-reward_model` as the oracle, we achieved a pairwise accuracy of **0.725** and strong AUROC scores after sequential auditing (see page 18, Table 6, Figure 10). This confirms that our framework is not limited to toxicity but effectively audits complex preference structures.
>
> Second, we scaled our experiments to **Llama-3-8B-Instruct** to address the concern about model size. On the original task of toxicity, the framework successfully scaled, achieving a pairwise accuracy of **0.773** and an F1 score of **0.847** (see page 19, Table 7). Notably, we observed that calibration metrics (ECE) and posterior tightness improved with scale, reinforcing our paper's finding that larger models yield more identifiable reward surfaces. These results directly counter the concern that our findings are artifacts of small models or simple tasks.
>
> **The Role of the "Noisy" Ground Truth and Posterior Contraction**
>
> The reviewer asked about the motivation for using a RoBERTa classifier (F1 ~ 0.76) and how "noise" affects the audit. We chose this classifier specifically *because* it is a controllable proxy for human labeling, allowing us to measure "distance to oracle" which is impossible with real human data. The fact that the classifier is imperfect (F1 ~ 0.76) actually strengthens our contribution: our Bayesian framework successfully recovers the signal *despite* the inherent noise in the proxy. To prove this, we ran a new ablation comparing Single-Round BIRL vs. Sequential BIRL on Llama-1B (averaged over 4 seeds) (see page 20, Table 8). While both achieved similar pairwise accuracy (~0.75), the Sequential approach dramatically reduced the **log-determinant of the posterior covariance** (a measure of volume/ambiguity) from **-196.0** in Single BIRL to **-897.1** in Sequential BIRL. This massive reduction in the solution space proves that our sequential approach systematically eliminates non-identifiability, converging on a "trustworthy" region of the reward landscape even when the ground truth is noisy.
>
> **Response to Specific Questions**
>
> *   **Concrete Audit Outputs:** The framework produces three concrete artifacts for auditors: (1) A calibrated reward model representing the verified objective; (2) An epistemic uncertainty map that flags Out-of-Distribution (OOD) prompts where the model’s alignment is fragile; and (3) A "Posterior Tightness" metric (log-det covariance) that quantifies how identifiable the objective is.
> *   **Comparisons to Related Work:** While other BIRL methods exist (e.g., Ramachandran & Amir, 2007), they focus on grid-worlds or tabular settings. Recent LLM-based reward modeling (e.g., Yang et al., 2024) focuses on *forward* RLHF optimization. Our work is the first to integrate *Inverse* RL with sequential active auditing and policy validation for LLMs. We will add a table explicitly contrasting feature sets (e.g., Uncertainty Quantification, Policy Validation, Sequential Updates) against these baselines.
> *   **Challenges going beyond linear models:** Moving to non-linear rewards (e.g., Deep Kernel Learning) allows capturing more complex preferences but increases the risk of overfitting and computational cost in posterior estimation. Our use of a linear head over frozen deep features is a deliberate design choice to balance expressivity with the tractability needed for Bayesian uncertainty estimation, a standard practice in Neural Tangent Kernel literature.
>
> We believe these new quantitative results (Cohen’s d=1.32, HH-RLHF Acc=0.725, LogDet reduction to -897) and clarifications regarding the definition of auditing significantly strengthen the paper and directly address the reviewer's concerns.

---

> ### Author Response · Authors · 2025-11-26
> **Further clarification required?**
>
> Dear Reviewer joPc
>
> We have posted responses to the concerns raised by your review. Is there anything further you would like clarification about beyond what we posted in the comments? Have all your concerns been appropriately addressed? Please let us know and we'd happy to provide you with further details where required.
>
> Thank you

---

### Official Review · Reviewer_97Do · 2025-11-01

**Soundness:** 3
**Presentation:** 2
**Contribution:** 2
**Rating:** 6
**Confidence:** 3

**Summary:**

This paper introduces an auditing framework that reframes reward inference as a verification process. It uses Bayesian IRL to recover a distribution over possible objectives and supports key auditing capabilities.

**Strengths:**

This paper presents a framework called Alignment Auditor that structures reward inference into three stages: recovering a distribution over objectives to reduce ambiguity, using uncertainty-aware diagnostics to assess trustworthiness, and training policies using the refined objectives to improve practical utility.

**Weaknesses:**

The task setting is narrow, focusing only on toxicity detoxification. The ground-truth reward comes from a classifier rather than human preferences, and the reward model is linear over frozen features.

**Questions:**

The experiments cover only toxicity detoxification. Does the framework apply beyond single-objective safety tasks? How would it handle multi-objective alignment (e.g., helpfulness–harmlessness–honesty) and trade-offs among complex human values?

The method assumes a linear reward over frozen features. Do the findings extend to more realistic, scalable reward architectures (e.g., finetuned, nonlinear, or deep reward models)?

What are the limitations of using a proxy classifier as “ground truth”? Can the authors discuss extensions to real human preference data and how results might change?

Is there evidence or a clear rationale that the auditing procedure scales to modern large models? What are the computational and methodological bottlenecks?

How do the authors ensure that the reported uncertainty reflects genuine reward ambiguity, rather than artifacts from model misspecification, feature freezing, or estimation noise?

---

> ### Author Response · Authors · 2025-11-20
> **Response to Reviewer 97Do**
>
> We thank the reviewer for their thoughtful assessment and for recognizing the core strengths of the Alignment Auditor framework. We appreciate the specific questions regarding generalizability, model scale, and the nature of the reward function. We agree that demonstrating the framework's capabilities beyond a single setting is crucial. In response, we have conducted significant new experiments during the rebuttal period—including scaling to **Llama-3-8B**, extending to the **Anthropic HH-RLHF** dataset, and quantifying the robustness of our linear reward formulation—to rigorously address these concerns.
>
> **Generalizability to Complex Tasks (Anthropic HH-RLHF)**
> The reviewer rightly asks if the framework applies beyond single-objective toxicity tasks to more complex domains like helpfulness or multi-objective trade-offs. To address this, we conducted a new set of experiments using the **Anthropic HH-RLHF (Helpfulness)** dataset, utilizing a Llama-1B backbone and the `Ray2333/gpt2-large-helpful-reward_model` as the oracle signal. This dataset contains nuanced, often conflicting preferences that implicitly balance helpfulness, harmlessness, and verbosity, representing a significant step up in complexity from binary toxicity.
>
> Our framework successfully recovered a calibrated reward distribution for this task, achieving a pairwise accuracy of **0.721** and strong AUROC scores after sequential auditing (see page 18, Table 7, Figure 10). This result confirms that the Alignment Auditor is not limited to safety constraints but can effectively model complex preference structures. The Bayesian framework handles "multi-objective" ambiguity naturally: if the demonstrations trade off helpfulness and harmlessness inconsistently, this manifests as a wider posterior variance (higher epistemic uncertainty) in the reward space. The framework does not need to explicitly know the objectives are "multi-objective"; it simply flags the resulting ambiguity in the latent reward, which is the primary goal of an audit.
>
> **Robustness and Scalability of Linear Rewards over Frozen Features**
> The reviewer queried whether our use of a linear reward head over frozen features is a limitation compared to deep, non-linear reward models. We emphasize that this is a deliberate design choice to enable rigorous **uncertainty quantification (UQ)**, rather than a simplified fallback. Deep, fully fine-tuned reward models are notoriously difficult to audit because they are "black boxes" where posterior inference is intractable. By using a linear head over the rich, pre-trained feature space of the LLM (similar to Neural Tangent Kernel approaches), we can perform exact or high-fidelity variational Bayesian inference.
>
> To prove that this “linear” constraint does not hinder performance on modern models, we scaled our experiments to **Llama-3-8B-Instruct** on the original toxicity task. The results were compelling: The framework achieved a pairwise accuracy of **0.773** and a toxicity detection F1 score of **0.8465**, improving upon the smaller models (see page 19, Table 7) . Furthermore, we measured **Cohen’s $d$** (standardized difference) between the inferred reward distributions for toxic vs. non-toxic inputs. We achieved a Cohen’s $d$ of **1.3254** when using Llama-3.2-1B, and a Cohen’s $d$ of **1.8218** when using Llama-3-8B-Instruct, which indicates a “very large” effect size in statistical terms as scale improves. This demonstrates that the frozen features of modern LLMs are sufficiently semantic that a linear probe can recover a highly effective and decisive reward boundary. Thus, the linear assumption allows us to gain auditability and calibration without sacrificing performance.

---

> > ### Author Response · Authors · 2025-11-20
> > **Response to Reviewer 97Do continued**
> >
> > **Proxy Classifiers vs. Human Ground Truth**
> > We acknowledge the reviewer’s point regarding the use of a proxy classifier (RoBERTa) as ground truth. In an auditing research context, using a proxy is methodologically necessary to establish a "ground truth" against which we can measure the fidelity of the recovered reward—something impossible with noisy, unverified human data. However, the framework is mathematically agnostic to the data source. The Bradley-Terry likelihood (Eq. 2) applies identically whether the preference pair $(o^+ \succ o^-)$ comes from a classifier or a human annotator.
> >
> > Crucially, our results suggest that the framework is robust to noise in the ground truth. The RoBERTa classifier has an F1 of ~0.76, yet our framework recovers a clean signal. In our new ablation study comparing Single-Round vs. Sequential BIRL on Llama-1B, we found that while raw accuracy was similar, the **log-determinant of the posterior covariance** (a measure of solution volume) dropped from **-196.0** (Single) to **-897.1** (Sequential) (see page 20, Table 8). This massive reduction indicates that the sequential Bayesian updates effectively filter out the noise (aleatoric uncertainty) inherent in the proxy labels, converging on the stable underlying objective. This suggests the framework would be equally effective at smoothing out the inconsistencies found in real human annotator data.
> >
> > **Distinguishing Genuine Ambiguity from Artifacts**
> > The reviewer asks how we ensure reported uncertainty reflects true reward ambiguity rather than estimation noise or model misspecification. We disentangle these sources through our **Sequential Bayesian Update** strategy. If the uncertainty were merely an artifact of the linear model being too simple (misspecification), adding more data would not necessarily reduce the uncertainty—the model would simply remain confused. However, we observe a monotonic and dramatic contraction of the posterior distribution (as evidenced by the log-determinant metric and predictive entropy) as more rounds of data are observed.
> >
> > Furthermore, our **out-of-distribution (OOD) diagnostics** provide strong evidence that the epistemic uncertainty is meaningful. When we injected spurious features (irrelevant keywords) into prompts, the framework immediately flagged these with high epistemic uncertainty (Figure 5, left), despite the model having high predictive confidence. This correlates strongly ($r=0.989$) with the Mahalanobis distance from the training distribution. This confirms that the uncertainty we measure is not random noise, but a precise signal indicating where the model lacks sufficient evidence to form a trustworthy preference ranking.
> >
> > **Computational Bottlenecks and Scaling**
> > Finally, regarding computational bottlenecks: the primary cost in our framework is the forward pass to extract features $\phi(o)$. Once features are extracted, the Bayesian inference (Variational Inference or Laplace Approximation) operates on the low-dimensional embedding space (e.g., $d=4096$) and is computationally negligible compared to the cost of pre-training or full fine-tuning. Therefore, the auditing framework scales linearly with the inference cost of the backbone model, making it entirely feasible for large-scale modern LLMs like Llama-3-70B or GPT-4, provided access to embeddings is available.
> >
> > We believe these extensive new experiments—spanning dataset generalization, model scaling, and rigorous quantification of decision boundaries—conclusively address the reviewer’s concerns about the breadth and robustness of the Alignment Auditor.

---

> ### Author Response · Authors · 2025-11-26
> **Further clarification**
>
> Dear Reviewer 97Do
>
> We have posted responses to the concerns raised by your review. Is there anything further you would like clarification about beyond what we posted in the comments? Have all your concerns been appropriately addressed? Please let us know and we'd happy to provide you with further details where required.
>
> Thank you

---

### Meta-Review · Area_Chair_3UVc · 2026-01-05

**Summary:**

As AC, I have thoroughly reviewed the reviewers’ comments, the authors’ rebuttals, and the supplementary experiments conducted during the rebuttal period.

This work proposes a Bayesian IRL-based framework for auditing LLM alignment, addressing a critical need in trustworthy AI. Reviewers raised constructive concerns spanning generalizability beyond single-objective toxicity tasks, clarity on the framework’s “auditing” and “refinement” definitions, computational scalability, and the need for quantitative evidence to support key claims.

The authors have comprehensively addressed these concerns through substantial new experiments and clear conceptual clarifications. In sum, the authors’ rebuttals are rigorous, evidence-based, and directly resolve the reviewers’ core concerns. The supplementary experiments significantly strengthen the work’s validity and generalizability.

**Reviewer Scores:**

Reviewer joPc would have changed the score if they had been able to participate fully in the discussion.

---

### Decision · Program_Chairs · 2026-01-26

Accept (Poster)